# "Us versus Them;" Local Social Fragmentation and Its Potential Effects on Building Pathways to Adapting to Wildfire

Mark Billings [1,*], Matthew Carroll [1], Travis Paveglio [2] and Kara Whitman [1]

1   School of the Environment, Washington State University, Pullman, WA 99164, USA; carroll@wsu.edu (M.C.); kmwhitman@wsu.edu (K.W.)
2   Department of Natural Resources and Society, University of Idaho, 875 Perimeter Drive MS 1134, Moscow, ID 83844, USA; tpaveglio@uidaho.edu
*   Correspondence: mark.billings@wsu.edu

**Abstract:** As the need for wildfire adaptation for human populations in the wildland-urban interface (WUI) intensifies in the face of changes that have increased the number of wildfires that exceed 100 thousand acres, it is becoming more important to come to a better understanding of social complexity on the WUI landscape. It is just as important to further our understanding of the social characteristics of the individual human settlements that inhabit that landscape and attempt to craft strategies to improve wildfire adaptation that are commensurate with local values, management preferences, and local capabilities. The case study research presented in this article evaluates social characteristics present in a WUI community that faces extreme wildfire risk to both people and property. It explores social processes that impede the ability of community members to work together collectively to solve problems (e.g., wildfire risk) and offers an alternative perspective about the nature of residency status (i.e., full-time and non-full-time) and its role in influencing wildfire mitigation efforts. This article closes with recommendations intended to facilitate collective action and foster community development.

**Keywords:** wildfire; wildland-urban interface; social fragmentation; social cohesion; adaptation

## 1. Introduction

The need to understand the social complexity of human populations in the wildland-urban interface (WUI) as it relates to wildfire risk has been well established [1–4]. However, it may not be enough to develop an understanding of particular human WUI populations at a specific point in time as it has been noted that the social characteristics of those populations are not static [5]. Previous research that focused on the effects of the increasing numbers and diversity of residents in WUI areas on wildfire risk [6] would suggest a need to generate a better understanding of how demographic shifts may contribute to shifts in social characteristics of WUI populations.

Beyond the change in the number of people who inhabit WUI areas, the effects of significant demographic changes may impact numerous social characteristics that influence the ability of those areas to live with wildfire. For example, a portion of new WUI inhabitants come in the form of amenity migrants who utilize the WUI as a secondary or 'home away from home'. It is well documented that, for many reasons, primary and secondary homeowners often have different reasons for being on the landscape and therefore different ideas about appropriate actions to reduce risk [7,8]. Additionally, even if there is agreement on mitigation strategies, the two homeowner groups may encounter different barriers in completing tasks that will lower their collective and individual risk [7,9]. Those differences have been found to lead to tensions between homeowner types [10,11]. This paper examines a process in which the ability of a community to coalesce around shared strategies to reduce wildfire risk is diminished. Our exploration of that process goes beyond a simple delineation between homeowner types (i.e., primary and secondary). This case study offers

empirically based insights into how differences, exacerbated by WUI emigration and shifts in local social characteristics, can impede strategies to reduce wildfire risk.

The fostering of fire adapted communities (FAC) is widely seen as an appropriate response to increased wildfire risk to human populations in the WUI [12–14]. Fire adaptation is a process which takes local social and biophysical context into account and is broadly seen as a population's ability to adjust to changing levels of wildfire risk [1,15]. Inter- and intra-community conflict can disrupt a community's ability to become fire adapted [16]. Exploring and developing potential pathways (i.e., a tailored combination of policies, coordinated actions, or incentives) that move a community toward fire adaptation has recently become a significant focus of researchers studying fire adaptation [13]. This literature has identified social roadblocks or barriers that are often imbedded within local social contexts that can hamper those efforts [1]. This paper addresses a barrier that has received relatively little formal attention in the wildfire social science literature, localized social fragmentation. Social fragmentation is a process by which a social group (e.g., community) has developed divisions among its members in such a way that those divisions negatively influence social mechanisms and may hamper their ability to work together to solve collective problems.

It has been well documented that locally based risk mitigation is a key to decreasing the risk of wildfire to human lives and structures [2,17]. It therefore follows that the public, land/risk managers, and policy makers must gain a better understanding of the local social contexts of WUI communities [3,5,18]. Recent literature and experience have strongly indicated that not all WUI communities are the same. Rather, it has been found that residents of various WUI communities are often characterized by a diverse set of preferences, abilities, and limitations. The challenge in all of this is that approaches to fire mitigation in local areas, if they are to be successful, must take account of these localized factors [5]. The identification and understanding of those localized factors can lead to the formation and implementation of strategies that allow particular WUI communities to adapt to and successfully live with wildfire. This paper extends on research to identify local circumstances and social characteristics that act as barriers to effective community adaptation to wildfire risk [1].

As the demography of the WUI has changed, so has the probability of large-scale wildfires in western North America. Recent examples of loss due to extreme fire events include the 2021 Dixie Fire which burnt over 740,000 acres and destroyed more than 1200 structures, and the 2018 Camp Fire that affected Paradise, CA. More than 18 thousand structures and 85 lives were lost [19,20]. The 2017 Tubbs Fire in California destroyed over 5000 structures and more than 2800 homes in Santa Rosa, CA [21]. The reasons for larger (i.e., more acreage burnt) wildfires include past land and wildfire management decisions which have increased the amount of fuel, coupled with longer and drier fire seasons due to climate change [6]. The east slope of the Cascade Range in Washington State has not escaped the trend of increasing wildfire risk to WUI populations [22]. In 2014, the Carlton Complex Fires destroyed almost 100 homes in just one town within this area [23]. A large portion of the east slope of the Cascades is located within an area that previous research led by the U.S. Forest Service has identified as the Wenatchee fireshed and as being prone to large fire events [24]. The Wenatchee fireshed contains numerous human settlements.

One particular residential area within the Wenatchee fireshed is of particular concern to land and risk managers due to its juxtaposition with highly flammable forest vegetation on all sides, and especially within the settlement area itself, and the fact that the residents have only one road by which to escape the area. These conditions have developed over many decades. The Ponderosa Community Club was incorporated in 1968 as a private campground in which individuals could purchase small lots to pitch tents and park camp trailers. Now, there are barely any lots in which a single-family dwelling has not been erected and an increasing number of those former 'vacation' homes are being occupied on a full-time basis. Their occupants are a mixture of former part-time residents who have retired to the community, amenity migrants, and people who work in areas near the community. This demographic shift has divided the community into two main groups, full-time and

non-full-time residents. An initial investigation in the case study area discovered potential discord between the two groups related to wildfire risk reduction efforts and prompted researchers to explore the potential differences between the groups and determine if those differences presented any barriers to collective action in the community.

Utilizing the Ponderosa case study, this paper aims to contribute to the understanding of social fragmentation within the context of wildfire adaptation in WUI communities. We explore the similarities and differences between the two main resident groups in this study area that share a common threat but see their local situation as "us versus them". We used an existing conceptual framework to explore social fragmentation within the context of this community and how it poses potential barriers that can limit pathways toward adaptation. Finally, we suggest approaches that might be used to overcome this challenge and allow the community to move forward with strategies to mitigate what appears to be a very significant source of risk to the community.

## 2. Literature

While the number of wildfire starts has remained relatively constant over the last two decades, the intensity of those wildfires has increased across western North America [12,25]. The 1990s saw an average of 78,587 wildfires in the U.S. that burnt 3.3 million acres. During the decade ending in 2020, the U.S. has experienced an average of 64,047 wildfires that has burnt over 6.8 million acres [22]. Of those national averages, the western U.S. (AK, AZ, CA, CO, ID, MT, NM, NV, OR, UT, WA, & WY) accounts for over 75 percent of acreage burnt during the time period 2015–2020, on average [26]. The cost ($279/acre) to suppress wildfires, adjusted for inflation, has remained relatively constant over the last three decades. Bigger wildfires (i.e., acres burnt) have increased overall suppression costs more than two and a half times to a point where the federal government is spending an average of 1.9 billion dollars per year [22].

In the period of 2016–2020, wildfires destroyed more than 43,000 structures in the western United States. More than 60 percent of those structures were homes [26]. Pyne [27] states that suppression, or as he puts it, a "policy of resistance", by itself has failed. The need for government agencies and WUI landowners to adopt strategies in addition to suppression to address growing wildfire risk is increasingly clear [12,25,28].

Federal agencies have employed various strategies other than suppression to combat the mounting risk that wildfire poses to resources and human populations in the WUI [29]. Examples of those efforts include the Healthy Forest Restoration Act which directed communities to create Community Wildfire Protection Plans (CWPP). These broad scale programs were intended to encourage communities to take the necessary steps (e.g., vegetation removal and evacuation planning) to prepare for wildfire [30]. Additionally, a voluntary program, known as Firewise USA, that promotes preparation efforts such as hazardous fuels removal around homes was implemented [31]. The U.S. Forest Service has created educational and messaging campaigns to reduce the number of human ignitions on public lands (e.g., Smokey Bear) and to educate children about the role of fire on the landscape (e.g., FireWorks) [32,33]. Some would argue that the Smokey Bear campaign has been incredibly successful at gaining public support for a fire suppression focus by messaging that emphasizes the destructive capacity of wildfire and the need for its removal from ecosystems. This success has had the unintended consequence of increasing the risk of large fires by increasing fuel loads [34].

Literature addressing wildfire risk documents three major contributing factors to the increase in the magnitude of and damage caused by wildfires. These include altered forest stand conditions due to prior forest management practices (i.e., fire exclusion), increases in the number of people residing in the WUI, and changing climate [6,25]. Social science literature concerned with wildfire risk has focused on modifying behaviors, understanding risk perception, and searching for ways to increase the ability of human populations to respond to wildfire events, to name a few [12,25,35,36]. There have been intense widespread calls to explore the concept of adaptive capacity of WUI communities [12–14] as a means of

increasing the capacity of human populations to adapt to a growing presence of wildfire. Adaptive capacity in this context is conceptualized as a combination of internal and external social and ecological factors in human communities that either promote or degrade the ability to prepare for, recover from, or adjust to future, previous, or current wildfire events [1,14,37]. Central to the concept of adaptive capacity is the ability of people to work together to address collective problems or threats [13,38,39].

Case study research has revealed that each WUI community has its own set of social characteristics that are influenced by the biophysical setting, the demographic characteristics of its residents, and the ongoing social interactions that take place in that locality. This local social context, in turn, influences the manner in which a community is able and likely to respond to challenges (i.e., wildfires) that face it. Utilizing an interactional approach for understanding community adaptation to wildfire (hereafter referred to as the interactional approach), Paveglio et al. [5] utilized characteristics they previously identified [4] to document similarities and differences between WUI communities related to wildfire risk adaptation. Those adaptive capacity characteristics (ACC) are instrumental in determining how communities respond to wildfire risk [1,5,13]. They are representative of traits within a community that are observable and bolster the capabilities of WUI communities to adapt to wildfire and also influence preferences related to actions residents of those communities are willing and able to take to reduce wildfire risk [1,5]. In other words, the interactional approach and ACC assist in synthesizing the essence of the social attributes of a given community in a manner that is germane to the adaptive capacity of that community and to reduce its fire risk by identifying barriers to and potential pathways for reducing risk.

A discussion of adaptation in the WUI must begin with an acknowledgement of social complexity resulting from the many differences among WUI communities located across a diverse biophysical landscape [18,37,39]. Local social context is dependent on the various social, biophysical, and historical characteristics that are specific to a particular community [40]. Adding to the complexity is the fact that those characteristics are not stagnant, but rather are in a near-constant state of flux [14,23].

Ongoing processes such as demographic shifts due to amenity migration, changes in resource utilization, and disparities among the historical experiences of local groups on the landscape come with a host of wildfire adaptation implications for WUI communities [23]. Amenity migration, for example, generally increases the overall population density of WUI landscapes. As it does so, it also shifts the overall demographic composition of communities in those locales as, on the one hand, property is often subdivided and land value increases while, on the other hand, the proportion of landowners with local ecological knowledge built up over years, or even generations, of living in a place decreases [41]. An influx of new residents to a community of long-time locals can result in social disconnection due to a lack of familiarity between new and long-time community members which can, in turn, affect the transfer of place-based knowledge about wildfire risk [41,42].

Experiences with hazard events and the potential of impending events can cause tension and conflict among community members [11]. Additionally, new community members may bring with them views that conflict with those of long-time locals about what constitutes a "healthy forest" or landscape [11]. In summary, a significant shift in demographics due to an influx of new residents can have cascading effects in a community that, in the end, may affect the ability of residents to work together to solve collective problems [38]. Elements of and changes in social context such as described here influence how effectively a community can work together [15].

As demographic shifts occur, the proportions of full-time and non-full-time residents may change as well. There are well documented differences in how each resident group perceives the use value of their home. Due to their residency status, full-time residents may be more dependent on the area's economy and have more and stronger social ties, while non-full-time residents who purchase homes for recreational purposes will be less concerned about the economic value of their home and have a stronger desire to maintain or improve environmental quality in the area [8]. There is an extensive amount of literature

pertaining to the differences between full-time (i.e., primary or permanent residents) and non-full-time (i.e., secondary or seasonal residents) [7–9,43].

For full-time residents that are retirees, they may have more time to conduct mitigation efforts on their property but may not have the financial resources to complete work that they are physically unable to do [9]. Full-time residents see a greater degree of efficacy of mitigation efforts than non-full-time residents, as well [7]. Non-full-time residents are more likely to cite a lack of time as one of the primary obstacles to completing mitigation efforts [7]. Additionally, non-full-time residents are more reluctant to cut down trees for fear of damaging the aesthetic quality of their residence and are generally less likely to engage in fuels mitigation efforts [38].

The literature also indicates that there may be differences in how full-time and non-full-time residents perceive risk of wildfire in their locale. Vogt and Cindrity [44] detected differences between the two groups related to the level of concern about wildfire risk when purchasing homes. Martin et al. [43] found, in a comparison between the two resident groups, that higher risk perception lead to increased fuel mitigation and that full-time residents were more likely to complete mitigation efforts.

*2.1. Pathways and Barriers*

The importance of understanding local social context, adaptive capacity, and organization/risk management in relation to scale becomes clear when attempting to determine which strategies or pathways are appropriate for residents of a given community to reduce wildfire risk. Pathways are loosely defined as a tailored combination of policies, actions, or incentives that are tailored to local, place-based conditions, and implemented to reduce wildfire risk [13]. Wyborn et al. [39] conceptualize pathways as an ongoing process that is informed and influenced by past actions and that redefines future possibilities. A pathway that works well in one type of community may not work in another. Research on social context and specific pathway approaches demonstrated, for example, that communities that are traditionally resource-dependent or working landscape communities [5] may have a better chance of reducing wildfire risk if they were given funds to self-organize volunteer firefighting and hazard mitigation capacity while a different type of community, formalized subdivisions, may require hired contractors or professionals to carry out these functions. Similarly, educational/outreach programs couched in formal scientific language may resonate among residents of a formal subdivision while messages framed in terms of the practical experience of longtime practitioners may find greater receptivity among more rural residents [1,13,40]. Just as each community type has pathways that have a better chance of succeeding, they also have barriers to success that are unique to their community type. For example, some communities are more trusting of outside agencies while some are distrustful of the same [1].

Understanding the adaptive capacity of human populations at the scale of 'community', as discussed above, is an example of the utility of considering scale as a function of the assessment of social characteristics of WUI populations. An additional use of scale that is applicable to adaptive capacity is the scale at which strategies will be implemented. Previous research has indicated that scale does matter in terms of which strategies, aimed at fostering adaptive behaviors, are implemented, and that scale is associated with the local social context of the community in question [38,45]. Previous research has identified several formal and informal organizational or risk management levels at which strategy implementation can take place: (1) National/State, (2) County/Local, and (3) Community-led [38]. Some researchers, stakeholders, and policymakers have advocated for increased adaptation across the entire WUI (i.e., national/state) [12]. The assumption underlying this research is that the lowest of the three levels, community-led, is the most crucial and least understood. Examples of community-led governance include but are not limited to participation in voluntary programs such as Firewise, and informal practices such as peer pressure to ensure fuels on private property are adequately managed [38]. Those strategies have been identified as having meaningful and enduring impacts on adaptation

at small, localized scales (e.g., community) [38,40]. Assessment of social characteristics and implementation of strategies designed to increase adaptive capacity of WUI populations is inherently interconnected with local social context and scale. Interactions across scales can also affect or influence adaptive capacity in WUI populations. For instance, as noted previously, unwelcome intrusion at the local (i.e., small scale) level by strategies that target entire regions and fail to account for localized conditions and experience can generate "pushback" and work against effective collective action at the community level [39]. Increasing the adaptive capacity of a WUI population has been linked to collective action and the establishment of collaborative risk initiatives which illustrates the need for productive dialogue that is tailored to the appropriate scale [4].

### 2.2. Social Fragmentation in WUI Communities as a Barrier to Successful Adaptation

The term fragment and all of its variants are widely utilized throughout wildfire literature. Fragmentation is sometimes used to describe the diversity of property ownership in the WUI [28,35]. More commonly, authors utilize the term in the context of habitat and vegetation "fragments" across a landscape [23,37,46]. Less frequently, fragmentation is utilized to describe the separation of people by physical or geographic barriers which culminate in disparate communities [11,47]. In all of these cases, the authors are attempting to communicate that there was something that was whole or unified at one point and has now been put asunder by biophysical or social forces.

Of primary interest in this analysis is fragmentation as it applies to social processes. Social fragmentation in the context of wildfire adaptation in the WUI has not been adequately addressed in the literature and when it is used, it is rarely conceptualized. Paveglio et al. [48,49] describe social fragmentation as differences in social characteristics such as values, skills, viewpoints, and connections to the land which shape or influence the division between communities. These authors suggest that fragmentation can impede successful collective action because actors may disagree on the optimal pathway(s) forward or even on the nature of the problem being addressed. Although the authors acknowledge the presence of social fragmentation at smaller scales, their focus has been largely heretofore aimed at describing inter-community divisions as they relate to finding agreement on pathways forward to reduce fire risk at a landscape scale [49].

We build from the Paveglio et al. [49] conceptualization of social fragmentation in the context of wildfire adaptation, as a social phenomenon that is exhibited as a social characteristic and/or a process in which a social group, community, or set of geographically connected communities exhibits an ongoing degradation or lack of social cohesion such that collective problem-solving is impeded [50,51]. One practical, adverse impact of social fragmentation related to wildfire risk reduction is a decrease in fuel removal. It is widely accepted that there has been a buildup of hazardous fuels throughout the WUI and that increased fuel loading increases risk to human settlements [25]. For instance, a commonly cited and recommended strategy to increase adaptive capacity in WUI communities is hazardous fuels reduction on public and privately owned land [46,52]. A lack of social contact between community members degrades social well-being among individual members which, in turn, can have several adverse effects including diminished willingness to support collaborative efforts that are necessary to complete effective fuels reduction programs [40,53]. As detailed previously, differences between groups with differing residency statuses (i.e., full-time and non-full-time) include divergent feelings regarding the values they place on the landscape [8] (Green et al., 1996). Differences in values is a social characteristic that is highly likely to influence social fragmentation [40].

Our conceptualization of social fragmentation shifts the focus from a large scale to focus on social fragmentation that is occurring at smaller scales (e.g., community and neighborhood). Additional processes that can influence social fragmentation include isolation of social group members or a diminished capacity within a group to provide support through social networks [54]. Conflict (inter-community and intra-community) can also foment social fragmentation, especially after hazard events that have deleterious effects

on the communities in the affected landscape [11,47]. Changes in demographic structure within a community or settlement area have been shown in some cases to exacerbate the above-described processes. Increases in community size or population density can shift social dynamics and weaken social ties [14]. Lastly, changes in or disputes over resource utilization, particularly on public lands, can create conflict between and within communities and further increase social fragmentation [23]. For all of these reasons, we posit that social fragmentation often represents a barrier to wildfire adaptation at the community and landscape scales [1].

The antithesis to social fragmentation as a process is social cohesion [11,14]. Paveglio et al. [48] refer to community development in the same context as social cohesion and situate community development as the polar opposite to social fragmentation as social characteristics within an area of interest. Increasing involvement (i.e., community development) within communities and between group members (i.e., social cohesion) can lead to more productive working relationships between community members and with personnel from outside agencies [55]. Social cohesion can be viewed as an item in a community's adaptive capacity toolkit that enables community members to reach agreement and act to decrease local wildfire risk [14]. Having a solid understanding of how all of the ACC are interrelated and that many social factors affect the ability of communities to address collective problems is fundamentally crucial to fostering adaptive capacity [5]. Being able to work on community-wide problems collectively is an important component of adaptive capacity because the community's safety is tied to each individual's action or lack thereof [14].

Literature that specifically addresses social fragmentation in relation to its effects regarding wildfire adaptation in WUI communities is sparse with few notable exceptions [13,40]. Paveglio et al. [13] present a useful description of how social fragmentation can affect wildfire adaptation by examining what role social dynamics play in forming collaborative units across landscapes that often consist of communities with different capabilities, values, and land management preferences. This is an important consideration given the fact that wildfire risk reduction takes place at scales that are often larger than one individual community but yet depends on actions at the community level. This paper expands on the findings of Paveglio et al. [13] by addressing the first piece of that larger collaborative puzzle, social fragmentation within an individual community. Specifically, this paper addresses the following questions:

1. What does social fragmentation look like at the community scale as it pertains to wildfire adaptation?
2. What possible strategies or pathways could be implemented to address social fragmentation at the very most local level in the WUI communities?

## 3. Methods

Data for this study were gathered and analyzed using an analytic inductive approach or what some refer to as negative case analysis [56]. One of the guiding principles of analytic induction is its reflexive nature where data gathered in the field influences the questions being asked about phenomena of interest and the analysis of those phenomena. This is similar to Glaser & Strauss's [57] grounded theory approach in that research questions and/or hypotheses are guided by what researchers are finding 'on the ground' and not by preconceived ideas about what should be found. In this case, the social phenomena being explored were barriers to wildfire adaptation. Researchers utilized focus groups and semi-structured interviews to collect data. Site-specific research questions were developed based on data collected from a key informant and two focus group sessions held with residents and land/risk managers. That process was used to guide data collection and analysis as categories were developed and either confirmed or disconfirmed as additional data was collected and analyzed [56].

Site selection was based on previous research that indicates that the east slope of the Cascade Range is susceptible to high risk of fire transmission in that geographic area [24,58].



The area (hereafter referred to as the Wenatchee fireshed) which included Chelan, Douglas, Kittitas, and Okanogan counties in Washington State was selected by members of the Co-Management of Fire Risk Transmission (CoMFRT) research group due to increasing risk to WUI communities within the area [59]. CoMFRT seeks to develop strategies to increase community resilience to wildfire by employing "deep dive" social science research within at-risk human populations and by working with land and risk managers within the at-risk area [59,60]. One of the human settlements at risk within the Wenatchee fireshed is the Ponderosa Community Club. The Wenatchee fireshed contains numerous human settlements which increases the possibility of significant loss of life and property in the event of large wildfire events [6,61]. A cursory visit to the area during the site selection process illuminated factors that would increase risk (e.g., limited ingress/egress and dense vegetation within close proximity to structures). Alternately, the community had recently been certified a Firewise community. Those attributes solidified the selection of the Ponderosa. Although there has been no wildfire activity within the borders of the community itself since its incorporation, community members have been evacuated due to wildfire activity on several occasions in recent years.

The Ponderosa Community is located on the east slopes in an area that has experienced destructive wildfires and is historically prone to stand replacement fires [10,62]. The wildfire risk faced in this area is being exacerbated by demographic changes within the community that decrease its ability to work collectively to address increasing risk. The Ponderosa is situated north of Leavenworth, WA, and the increasing wildfire risk that it faces is similar to that faced by other communities in the region and indeed in the wildland urban interface (WUI) (i.e., communities in which human settlements are interspersed with wildland vegetation that is undeveloped [6,63]). The Ponderosa sits on approximately 1.23 km$^2$ in which there are just over 600 individual parcels with 594 privately owned properties [64]. As we will detail below, this particular community faces a particularly daunting set of circumstances which include extensive proximity of closely-spaced homes and buildings to flammable vegetation, a single narrow road as the only route for possible evacuation, a population bifurcated by residential status (full-time residents versus non-full-time residents (i.e., those who use their residences as vacation homes)), and a belief by wildfire experts that the potential for a disastrous fire in the area is high and ever-increasing.

> "There is a glaring lack of fire history in the Chumstick drainage. Pretty much, this whole drainage hasn't burned in modern times. That risk just keeps transferring every fire season to the next season. At some point, our luck's going to run out here."—Wash. DNR Land Manager

Focus groups function as a practical method for capturing the views and preferences of a group of people, particularly from within a relatively small, bounded population. They often lead to discussion between group members that are rich in description, and they offer participants the opportunity to challenge each other's ideas, which fosters additional detail within responses. The exchanges between participants becomes as useful as the questions asked by researchers because the perspectives of other participants can encourage deeper reflection by all participants regarding their views about a particular topic [56,65].

Recruitment for focus groups followed an established strategy that has been used in similar case studies [1]. An online property search database managed by the county assessor in conjunction with other web-based applications (i.e., White Pages & Zabasearch) were used to determine the names, addresses, and phone numbers for all of the privately owned parcels within the community to create a sample frame. Recruitment of land/risk managers was completed by developing a comprehensive list of organizations (governmental and non-governmental) that have a stake in wildfire risk reduction in the area. Researchers attempted to have at least one representative from each of the identified organizations present in the professionals focus group. For example, researchers contacted U.S. Forest Service personnel responsible for the area along with Washington State Department of Natural Resources, area Extension agents, local fire department members, and representatives of Chelan County. The goal throughout the recruitment phases was to generate a

representative cross section of the community and people responsible for land and risk management of the area. To achieve that goal, the recruitment process continued until researchers were confident that the focus groups would be diverse in demographic and social characteristics.

Data from the first two focus groups indicated that there were two potentially distinct groups within the study site. The first group consisted of full-time residents (i.e., primary homeowners). The second group consisted of part-time residents (i.e., secondary homeowners). Data from full-time residents indicated a potential difference in perceptions related to wildfire risk reduction. The authors developed a plan that would appropriately study each group separately to determine similarities and/or differences between the two groups. The primary author utilized the original sample frame and divided it into two groups based on the presence or absence of a secondary address listed in the county assessor's database. Property owners with a secondary address outside the community were assigned to the non-full-time resident group and property owners without a secondary address were assigned to the full-time resident group. The majority of non-full-time residents' primary residences were located on the west side of the Cascade Range, in or near the Seattle metropolitan area. Those residents were recruited via phone to participate in two focus groups that were held near the Seattle area (one north of and one south of Seattle). Full-time residents were recruited for focus groups via phone calls and in-person solicitation within the community. That focus group was held in the Ponderosa Community. Snowball sampling was used to recruit full-time residents in cases where the resident could not be reached by phone and to create a more diverse set of focus group participants that held differing experiences or opinions within the community [56]. Upon initial contact with a potential participant, researchers confirmed residency status with the resident and gathered information about the length of time they had owned their property or lived full time in the community.

Focus groups were conducted in October 2018, June 2019, and September 2019. Researchers conducted a total of five focus groups with 52 participants (seven professionals, 23 non-full-time residents, 10 full-time residents, and a mixed residency group of 12). During that same time frame, researchers conducted semi-structured interviews with an additional 27 residents (17 non-full-time & 10 full-time). The primary purpose of conducting the additional interviews was to ensure theoretical saturation (the point at which no new themes are emerging from the data) [66]. All focus group sessions and interviews were audio recorded and later transcribed. Transcriptions from resident focus groups and interviews were anonymized by the primary author.

Analysis of the data occurred in several phases that moved from a broad to a narrower focus over the course of the process. The first phase consisted of a discussion between researchers about potential barriers to wildfire adaptation within the context of the study site. This led to the identification of potential themes that could be explored in the formal analysis of the data. The second phase of analysis consisted of a systematic process of analytic induction and thematic analysis. Thematic analysis requires multiple rounds of coding. In this study, topic and pattern coding were used during data analysis [67]. Participant perceptions about various topics were identified and organized into categories and then into themes [68]. Analytic induction assisted in the development of an understanding of social phenomena through a thorough examination of the data [69]. Researchers used a combination of manual and computer-assisted qualitative data analysis software (CAQDAS). QSR NVivo 11 software was used by the primary author while the second and third author coded manually. All three authors met regularly to discuss emerging codes, categories, and resulting themes from their individual efforts. Additionally, having multiple people code the data provided intercoder reliability.

The methods employed in this study fall within the accepted norms of qualitative research [65,68]. They provide a detailed description of the two groups that are of interest in this study and achieve theoretical saturation of the complex social characteristics that have an effect on each group's ability to become more fire adaptive [5]. The use of this

methodology produced lessons that are similar to other case studies that have used similar methodological approaches [1,5,13,15,37]. The lessons learned through the use of this methodology provide for transferability (i.e., theoretical generalizability) [70].

## 4. Results

A full-time resident offered an observation about non-full-time residents in the community saying, "Most of the people who live here, 80 percent of our people, this is a second home. This is a vacation home". When asked if that meant that non-full-time residents didn't have as much at stake in the community, they answered, "Yes". A non-full-time resident responded, "This is more my home than my [primary residence] and it would be very sad if my home here [were destroyed]. I've put sweat, blood, and tears in this home, and it means a lot to me". As we will describe below, further investigation indicated that this exchange was emblematic of a significant social division within the community that appears to be constraining the ability of the two local groups to work successfully together to reduce fire risk.

According to area land managers, the landscape surrounding the Ponderosa Community consists of a mixed-conifer forest dominated by a large ponderosa overstory. Land managers state that although the area is still somewhat representative of traditional forest composition, community residents have introduced species that have the effect of changing fire risk in the area. Residents have planted non-native species around homes which are more flammable than native species and have the effect of increasing risk in home ignition zones. Land managers also cite the increased presence of shade tolerant species (i.e., grand fir) that would not have been present prior to the development of the area and the suppression of wildfire that would have minimized the amount of ground and ladder fuels as factors that increase wildfire for the Ponderosa.

Land and risk managers that participated in the focus groups held the unanimous opinion that the community and the public lands adjacent to the community were hazardous due to the amount of fuel in the area. They were also in agreement about the additional risk created by abundant fuel loading within the community. As a representative from one agency put it, "When I drive in here . . . my neck stands up and I'm like, 'oh man, this is bad'".

The community was designated a Firewise community in 2014. The effort to receive that designation was led by a full-time resident. Under the leadership of that resident and in cooperation with the local conservation district, programs were put in place to reduce ground fuels such as pine needles and increase education about wildfire risk in the area. Prior to receiving their Firewise designation, community members obtained grant funding that was used to improve communally owned parcels and complete risk assessments on approximately 10 percent of the privately owned properties in the area. Fuel reduction projects in common areas focused on the removal of shade tolerant tree species, leaving only ponderosa pine remaining. Despite this effort, members of the local fire district that participated in our focus group expressed doubts about the possibility of successfully fighting a large fire should one occur in the community. In addition, several participants voiced concerns related to the number of structures in the area, the close proximity of those structures to each other, and the additional risk that is posed by that situation. The community's Firewise leader said, "We have 500 plus structures, and so I'm going to make everybody mad at me now, but in regard to Firewise, we could clear cut this place, and still the structures would be a problem".

Housing and vegetation density as well as limited ingress/egress were cited by firefighters who joined our focus groups as the three primary reasons for an unwillingness to send firefighters into the community in the event of a large wildfire. One local firefighter said, "From a firefighting point of view, if we ever had a crown fire in here, we probably wouldn't come in here". Participants representing land/risk management agreed that improvements could be made to reduce risk to the community, but in the words of one

participant, "There's a different mindset [in the Ponderosa] . . . these people are not really vocal". The possible reasons for this are explored below.

*Evidence of Community Changes*

According to participants, the Ponderosa is a community that has been slowly changing over the last 60 years. The original layout of the area was set up to accommodate a transient population of campers. In describing the area, a full-time resident said, "Initially, Ponderosa was designed to be just a weekend or vacation place. It really wasn't designed for houses". A non-full-time property owner who has been using the area for multiple decades commented, "When we built our cabin, there was two cabins on the whole area, 52 years ago". What began as a vacation and recreation location has transformed into something that, for some, is beginning to resemble the type of development that they were seeking to escape when they originally bought their place in the woods. A non-full-time participant explained, "It's becoming bigger, more people, less friendly, more and more expensive as more and more amenities are added. It's not as rustic as it used to be". The change, according to participants, has moved in a singular direction and toward a specific type of ownership over the years. A full-time resident said, "I think that some of the full-time people like to think of it as an old folks' retirement center, which it is not, and they get a little grumpy about people with kids". This perception is common among the majority of people who participated in this study.

The problem associated with this change, as described by participants, is an increasing sense by participants that there is an overall loss of community in the Ponderosa. "People are not seeing it as a community anymore. It used to be everybody waved at everybody, everybody was very polite. It's not like that now", said a non-full-time participant. The continuing shift has changed the overall makeup of the population. Fifty years ago, there were a handful of full-time residents. Now, approximately 30 percent of the property owners are living in the community on a regular basis and the shift is apparent to community members, as a full-time participant noted, "Even from my front porch, I look around and the 10 homes I can immediately see, there's only three of us that are year-round residents".

The two primary resident groups (i.e., full-time and non-full-time) have different reasons for owning property or living in the Ponderosa. Full-time residents were vocal about their enjoyment of the area because of the relative "peace" that living there affords them with full-time residents commenting that, "I like how quiet it is during the week", and, "I love the fact we're in the mountains. We're along the river. We have views. It's peaceful". In addition to living in a "natural" environment, this was the most common reason given by the full-time group for owning their property. Non-full-time residents, however, have different motivations for wanting to own and use property in the community. For the vast majority of the non-full-time residents, their primary motivation for having a place in the Ponderosa centered around the opportunity for recreation. A non-full-time resident said, "There's so much recreation over there, it's unbelievable. If there's some kind of sport you want to do, bicycling, climbing, you know, whitewater, whatever, hiking, backpacking, it's like heaven".

The bifurcation between the two groups is recognized by members of both, full-time and non-full-time residents and was cited by participants as a source of friction between the groups. As previously noted above, disruption of social cohesion can diminish collective action. One non-full-time resident summed it up best saying, "I kind of think they have an 'us vs. them' mentality". Although the non-full-time residents outnumber full-time residents more than two to one, they feel ostracized by their full-time counterparts with one participant lamenting, "So now that the people have decided to retire there and spend their lives there, I am considered a weekender and it just blows me away, the way I'm treated". This attitude was prevalent among non-full-time participants and was cited as a barrier to social cohesion between the two resident groups. Full-time resident participants' reasons for treating non-full-time residents in a way that reinforces these perceptions are grounded in their perception that non-full-time residents do not share their values as it

pertains to maintaining the area and protecting its natural beauty. A full-time participant said, "They don't seem to have as much care and appreciation . . . of the beauty in the place. They seem to bring a lot of noise and a lot of the crowdedness that's on the west side of the mountains". This quote illustrates another conflict that relates to one of the aforementioned reasons that full-time residents cite as a value for living in the Ponderosa, peace and tranquility.

In addition, full-time residents' perceptions, based on their observations, have led them to conclude that they have a greater affection for the area with a full-time participant asserting, "During the summer, it gets really crowded with tourists that don't really seem to understand the importance of the . . . and the benefits of being here in nature, and loving it as much as we do". Non-full-time residents, on the other hand, see full-time residents as not valuing the original intent of the community as a place for recreation.

> "They want to make it into a regular neighborhood. They want to urbanize it. I don't want it urbanized. I have an urban place in Seattle, that's why I have a place [in the Ponderosa] because I don't want urban."—Non-full-time Resident

Another difference between the two groups is in how connected they feel to the overall community. Full-time residents are more likely than non-full-time residents to perceive a higher level of connectedness to the community. A full-time resident said, "It's a nice little community, . . . fairly close knit". The connections that full-time residents have within the community tend to be with other residents that share their values and residency status. A full-time participant said, "I do like that it's a pretty intimate community. Meaning, the neighbors that we are close to, that are the other year-round residents, it's easy to walk to their house to visit, to fellowship, to hang out". Full-time residents have, due to the amount of time they are in the community physically, increased opportunities to form relationships with other residents.

> "We're pretty well tied in. We know a lot of people. We're not very far from the clubhouse. Often, we'll go in there and have one of the potlucks or whatever and see all the people and get together."—Full-time Resident

Although non-full-time residents may enjoy relationships with neighbors that are in close proximity, they generally stated that they do not feel a strong connection to the community. A common theme among the non-full-time residents is that they have limited opportunities, because of time restraints, to form connections with other community members. This, according to participants, is the primary reason they do not see themselves as fully enmeshed in the community as they may like. As one participant said, "I haven't really established relationships with people because really I'm part-time. I mean, I'm hardly ever there".

As it pertains specifically to issues related to wildfire, full-time and non-full-time residents are similar in several ways. Both groups expressed a recognition that there are physical limitations to what they can accomplish to reduce risk due to the size of their properties and the close proximity of their neighbors. A non-full-time resident stated that, "But what I know is, we're only as safe as our neighbor, just like the herd immunity".

Both groups share similar views pertaining to strategies that they believe will minimize risk to their properties. The presence or absence of pine needles is an indicator of property maintenance. A non-full-time resident expressed this by saying, "But the pine needles, I mean some people's yards, [six inches] deep with pine needles". In addition to pine needles, participants described a need to "limb up" trees and remove brush from around structures.

One area where the two groups differ in regard to wildfire issues is perceived risk. Non-full-time participants stated that they understand that there is inherent risk associated with owning a place "in the woods", but express their concern about wildfire risk in general terms. When asked if they thought they or their property was at risk from wildfire, most of the non-full-time participants responded in a similar fashion, "No more so than everyone else, but yes. The whole area is, I think, always at risk". Another non-full-time participant

said, "Not high risk, but yeah, I think everything on that side of the mountains is potentially, but not a high risk. I don't expect it to burn down, but you know".

Full-time participants expressed their concern about wildfire risk in more immediate terms with one respondent stating, "It's not an if, it's a when". Full-time residents also expressed concern over increased risk due to their geographic location and stated that the risk specifically impacted their properties. A full-time participant said, "I think it's high. I mean, I thought there was something that came out that our county, not just the Ponderosa, but that we are one of the highest risk areas in the state".

During focus group and interviews, both full-time and non-full-time residents frequently mentioned matters related to participation in social events, community planning and fuels mitigation efforts. Most participants acknowledged that activities like the removal of flammable vegetation is a sound strategy for reducing wildfire risk. However, full-time residents generally hold the perception that non-full-time residents are not adequately engaged in activities that improve community safety. A full-time participant said, "There's a lot of property owners in here that are not here all the time to take part in any of this stuff and we've already done it". In general, full-time residents associate the lack of participation in their counterparts with non-full-time residents having different priorities and a lack of concern about safety and their properties.

> "And the people that have bought property that just sit on it and don't do anything with it are the people that are what we call 'weekenders' that come in only to recreate and have fun, don't take time to clean their places up, that sort of thing."—Full-time Resident

Non-full-time participants were admittedly not as participatory as full-time residents, but their reasons for not participating in community events, property maintenance, and fuels removal were different than the reasons perceived by the full-time residents. Non-full-time respondents stated that one of the biggest barriers to completing necessary work around their property is, according to one participant, "time and access". Another stated that, "I'm not over there a lot and don't have the time to do it and I mostly do it. Just not being up there for enough days". Other barriers to participation that non-full-time residents cited included not being informed about events or activities, lacking financial resources, and not having a designated area to place removed material safely and conveniently.

An area that both resident groups appear to agree on (albeit for different reasons) is their dissatisfaction with the management of the community. Both full-time and non-full-time residents stated that they believe that there is a general lack of oversight within the community. One participant said, "There are no rules, and if there are, they aren't enforced". In most cases, participants were referring to property maintenance and development of new structures within the community. A full-time participant voiced this concern saying, "The board doesn't do anything about it. You're only supposed to have one living unit on your property and you can walk around here and see that there's a lot more than one living unit on the properties". Non-full-time participants also stated that they were displeased with how local leadership managed the community green spaces that are interspersed throughout the Ponderosa.

> "They have left behind piles of bad timber . . . that they didn't want. And nobody came in and cleaned up after them, and so yes, the risk, if lightning strikes there, there's nothing but a fire hazard waiting to happen."—Non-full-time Resident

Full-time residents expressed frustration over a different specific issue that came up repeatedly in focus groups and interviews, vacation rentals (i.e., air bed and breakfasts). Vacations rentals are seen by both groups as potential sources of human fire ignition. A full-time resident expressed this frustration, saying, "I think the thing that bothers most of us, the property owners and especially us, is the nightly rentals and the noise and the dangers they bring in". Participants credit a lack of oversight by the Ponderosa management and a lack of structure by the Board of Regents with creating a situation that they perceive as a disruption to the peace and tranquility of their community and a potential threat to

community safety due to the risk of human ignition by renters of vacation properties who may not be as familiar with local risks and regulations designed to minimize those risks.

## 5. Discussion

According to area land/risk managers, the Ponderosa Community faces extreme fire risk due to the amount of fuel (i.e., vegetation and structures) present in and around the community. While they have engaged in efforts to work with the community to reduce that risk, local land and risk managers would like to see residents of the Ponderosa much more engaged in efforts to reduce wildfire risk. Specifically, land/risk managers would like to see a greater concerted effort on the part of residents (full- and part-time) to reduce vegetation in the area. The Ponderosa is not alone in facing this situation and, as pointed out by one of the participating land/risk managers, is demographically similar to other settlements throughout the Wenatchee fireshed as it pertains to ever increasing wildfire risk. To adequately address wildfire risk in this community would require wholesale vegetation management projects which involves the cooperation of most, if not all, of its members. What makes the Ponderosa especially challenging is the inability of its residents to work together in addressing that risk.

We have built upon Paveglio et al. [13,40] to conceptualize social fragmentation as a process in which a group of people, a community, or a set of communities that once experienced social cohesion but has since been torn asunder by various social mechanisms. In the case of WUI communities, those mechanisms include, but are not limited to, demographic changes which diversify values and collective identity within or between communities [40]. Research pertaining to WUI population demographic changes indicates that the trend of people immigrating to rural areas that are at increasing risk of wildfire has remained constant [6]. For this reason, developing a better understanding of social fragmentation and its impacts on collective action is an important component of increasing adaptive capacity in WUI communities.

It is crucial to understand the role that social fragmentation plays in affecting intra-community collaboration and cooperation. As previously stated, there are a number of factors that can foment social fragmentation and the Ponderosa represents a prime example of those mechanisms at work. An increasing number of full-time residents in the Ponderosa has increased the number of people whose values have changed as such. Those values (i.e., the protection of their primary homes for use value and economic value) run counter to the values of non-full-time residents who see the Ponderosa as a vacation destination. This shift in how the two groups perceive what the location means to them has created a rift between the two and has created an "us versus them" mentality. At the time field work was completed, full-time and non-full-time residents held competing notions about what should be done to address collective problems with full-time residents favoring strategies to increase vegetation removal and non-full-time residents focusing attention on a lack of social connection to community activities. Interactions between the groups are rare and there is little evidence that any community members are actively attempting to span the divide. The lack of interaction between the two resident groups makes it difficult for them to recognize some of their shared values and concerns about the community.

Perhaps intuitively, blaming the large number of non-full-time residents (i.e., amenity migrants) for the shift in attitudes about how this community should be utilized and what it should mean to its members would fit with literature that suggests this as the norm [7,44]. However, it is important to remember what the Ponderosa Community Club was originally structured to be, and that the changes occurring related to social context and adaptive capacity are being driven by those residents who are choosing to live permanently in an area that was originally formed as a private campground, and for many non-full-time participants remains as such. The relevant takeaway in the Ponderosa is that changes have occurred, and the result of those changes has been the breaking apart of the whole, creating two disparate sides of a community which faces immense challenges.

Currently, according to participants, the community does not possess the organizational structure or competence to address wildfire risk reduction, nor to bring residents together in a way to reach agreement and buy-in on a way forward with fire risk mitigation. Participants from both resident groups expressed a distinct dissatisfaction with community leadership. Any risk reduction efforts that have taken place have largely been spearheaded by small groups or individual community members (e.g., Firewise). Unfortunately, the residents who have led those efforts have typically been from only one resident group, full-time residents, which has had the effect of increasing the rift between the two groups, reinforcing the "us versus them" mentality. This is unfortunate as it has been noted that one of the benefits of programs like Firewise can have the opposite effect and assist in community development [4].

It is important to draw a distinction between what some might perceive as community diversity and the ongoing process of social fragmentation in the Ponderosa. The data collected in this community suggest that there were, at one time, increased levels of social cohesion, but based on the results of this case study, it is clear that the Ponderosa is now socially fragmented and continues to suffer degradation of social cohesion as the process of social fragmentation plays out. That process has interrupted the ability of Ponderosa inhabitants to form community fields (see Wilkinson, 1991 for description) which influences their ability to work together collectively to reduce wildfire risk. This case study highlights the need to increase our understanding of social fragmentation as a process and how to recognize it as a social characteristic instead of social diversity when assessing appropriate pathways toward wildfire adaptation.

The Ponderosa is not unique in the problems that it faces due to social fragmentation as illustrated within other case studies (See Carroll et al. [11] for example). While the pathways for this community may not be universally acceptable for other similar communities, it does point to the need to understand each community and its social characteristics using a systematic approach (e.g., the Interactional Approach). Those social characteristics are often related to the scale at which strategies to address wildfire adaptation are applied. Scale is a crucially important yet understudied component of social fragmentation. The argument here being that while social fragmentation occurs at multiple scales [13] and creates a barrier to adaptive pathways, addressing social fragmentation at the scale at which it occurs may produce the greatest results. If social fragmentation is occurring at the local scale, it would be difficult to achieve meaningful and lasting change across the landscape. As Paveglio et al. [13] noted, it is often divergent values or worldviews held by different groups of people that are likely to produce barriers to engaging in pathways toward wildfire adaptation.

Implementing strategies to reduce wildfire risk will first require processes that help residents of the area find common values or purpose and agree upon the commitments they will make to help address their shared circumstances. In other words, the community must experience some level of social cohesion. The task of bringing the two resident groups together will take a considerable amount of time and effort, but effective solutions are often long-term solutions that forego one-time fixes [39]. A lack of social cohesion could result in diminished adaptive capacity through a degradation of overall community well-being and trust among members [4,49]. A pathway for wildfire adaptation in the Ponderosa would consist of forging connections between the two resident groups. This will be challenging considering that full-time and non-full-time residents rarely interact and when they do occur, interactions are often contentious. For this reason, it may be necessary to build that interaction through a facilitated process by an outside, neutral, and impartial facilitator who is trained in the techniques and process design that can help multi-party groups collaboratively address challenging and complex issues.

This case study provides the beginning stages of a situation assessment, an important step in a conflict process designed to identify the parties involved, the complex relationships between those parties, the issues of concern, and potential barriers and opportunities (e.g., capacity, resources, etc.) for addressing those issues through a collaborative process [71,72].

In this case, those include issues surrounding differences in perceived values between the two groups, a perception of a lack of involvement by non-full-time residents by full-time residents, and a perception of a lack of access to community involvement by non-full-time residents. It may be appropriate in this case for one or more local or state agencies (e.g., Washington State Department of Natural Resources) to provide funding to bring in a skilled neutral facilitator. There is evidence from participants' and area land/risk managers' comments in the focus groups that the Ponderosa would be willing to accept assistance from outside the community. A facilitator would first determine if a collaborative process would be appropriate and then, if so, design a collaborative process that addresses the challenges of getting the two different resident groups and the local and state agencies that manage fire and fire risk in the Ponderosa region "in the same room".

If a collaborative group could be successfully convened, it would be important to next develop a shared common understanding of 'the problem', come to an agreement on the facts, which may require joint factfinding, explore each party's issues and interests, and identify shared values. This could entail, for example, instituting strategies to increase communication between the two resident groups–in other words, increase social cohesion by interrupting the process of social fragmentation. As mentioned in the results, non-full-time residents often feel left out of important communal decision-making processes. One participant suggested making community meetings available online (e.g., Zoom). Additionally, participants from both resident groups are troubled by the lack of egress options (i.e., one way out) during an emergency situation and both are dissatisfied with the management of vacation rental properties. These areas of common ground seem ripe for bringing members of the two groups together to work on collective problems because the identification of shared values interrupts the process of social fragmentation. It is important to note that these first stages will take time but could lead to improved social cohesion, allowing for all parties to work collaboratively to reach agreement and commitment on fire risk management.

The interaction that was documented early in the results section, between the full-time resident and the non-full-time resident regarding their disagreement over whether they hold similar values was the impetus for this study on social fragmentation. While the interaction was contentious, it also offered a glimmer of hope, in that both residents felt a deep appreciation for their place in the Ponderosa and the community itself. An outside facilitator should capitalize on that shared appreciation, even it is for often differing reasons.

**Author Contributions:** Conceptualization, M.B., M.C., T.P. and K.W.; methodology, M.B., T.P. and M.C.; validation, M.B., M.C. and T.P.; data collection, M.B. and M.C.; formal analysis, M.B.; writing—original draft preparation, M.B.; writing—review and editing, M.B., M.C., T.P. and K.W.; Supervision, M.C.; Project administration, M.C. and T.P.; Funding acquisition, M.C. and T.P. All authors have read and agreed to the published version of the manuscript.

**Funding:** This research was supported by the Rocky Mountain Research Station of the USDA Forest Service, United States (agreements 19-JV-11221636-073, 21-CS-11221636-080, 19-JV-11221636-083).

**Institutional Review Board Statement:** Ethical review and approval were waived for this study because the study would not reasonably place the subjects at risk of liability or socioeconomic damage in the event of a disclosure.

**Informed Consent Statement:** Informed consent was obtained from all subjects involved in the study.

**Acknowledgments:** Thank you to Eli Loftis for their assistance with coding and Jessica Billings for copy editing.

**Conflicts of Interest:** The authors declare no conflict of interest.

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
