# Peer review of "“Us versus Them;” Local Social Fragmentation and Its Potential Effects on Building Pathways to Adapting to Wildfire"

_fire, doi:10.3390/fire4040096_

Round 1

Reviewer 1 Report

Thank you for the opportunity to review this manuscript on social challenges in the wildland urban interface that make it difficult to apply land management strategies at a scale needed to match the extreme wildfire risk.

Overall, this manuscript offers valuable insights about the need to view WUI fire mitigation efforts at the neighborhood (or collective) level, rather than focusing on individual properties/owners. This conceptual move toward the neighborhood level is crucial given the large scale of WUI neighborhoods due to increased uses and the blend of ‘ecosystem services’ required in such areas.

A second valuable contribution of this study is the practical advice provided in the Discussion. The authors emphasize using a 3rd party to mediate conversations between factional groups (in this case full-time vs non-fulltime residents) in efforts to reach consensus on landscape priorities. Given these practical recommendations, I would recommend citing a few pieces that make a case for what is at stake when such efforts occur too late or not at all (especially near lines 266-270, and in paragraphs starting at line 784).

I noted a handful of typographical errors, so please conduct a thorough copy edit. Best of luck with this work!

Reviewer 2 Report

line 147:  It could be helpful to explain the context for why Smokey Bear is seen as "too successful". The increasing fuel loads is usually attributed to forest management and climate change, as you highlight in the previous section. The role of Smokey Bear is demonizing all fire or creating fear of all fire is more the issue. 

191: "local ecological knowledge" It may be warranted to discuss how relevant LEK is given the changing climate and the barriers it might create. Long-term residents may be less willing to change traditions to become more fire adapted as those traditions have worked for generations  

208: 2nd home owners who buy as an investment property would likely be very concerned with the economic value of their home.

300: May be good to present the counter belief as well here, that hazardous fuel reduction is not effective and harmful to other resources

503: The term overstocked with timber suggests a commodity approach to management, rather than explicitly wildfire risk reduction. This may be worth noting.

710: I suggest softening the language "lack of ability" .This may be an assumption. They have not worked together on this issue but they have also not had dedicated resources, leadership or felt urgency. Maybe they just lack motivation ? have other more pressing priorities? 

728: The values are not necessarily counter to each other as both groups values recreation, nature, etc...There is more overlap than conflict 

755: this is an interesting commentary on the Firewise program. It points to a real weaken in that program. Maybe that is worth highlighting?
